

# Population structure and diversification of *Gymnospermium kiangnanense*, a plant species with extremely small populations endemic to eastern China

Xiangnan Liu[1], Meizhen Wang[2], Shiqiang Song[1], Qing Ma[3] and Zhaoping Yang[1]

[1] College of Life Sciences and Technology, Tarim University, Alar, Xinjiang, China
[2] College of Life Sciences, Zhejiang University, Hangzhou, Zhejiang, China
[3] College of Biology and Environmental Engineering, Zhejiang Shuren University, Hangzhou, Zhejiang, China

Corresponding authors
Qing Ma, maqing90@live.cn
Zhaoping Yang, yzpzky@163.com

## ABSTRACT

**Background:** *Gymnospermium kiangnanense* is the only species distributed in the subtropical region within the spring ephemeral genus *Gymnospermium*. Extensive human exploitation and habitat destruction have resulted in a rapid shrink of *G. kiangnanense* populations. This study utilizes microsatellite markers to analyze the genetic diversity and structure and to deduce historical population events of extant populations of *G. kiangnanense*.

**Methods:** A total of 143 individuals from eight extant populations of *G. kiangnanense*, including two populations from Anhui Province and six populations from Zhejiang Province, were analyzed with using 21 pairs of microsatellite markers. Genetic diversity indices were calculated using Cervus, GENEPOP, GenALEX. Population structure was assessed using genetic distance (UPGMA), principal coordinate analysis (PCoA), Bayesian clustering method (STRUCTURE), and molecular variation analysis of variance (AMOVA). Population history events were inferred using DIYABC.

**Results:** The studied populations of *G. kiangnanense* exhibited a low level of genetic diversity ($He = 0.179$, $I = 0.286$), but a high degree of genetic differentiation ($F_{ST} = 0.521$). The mean value of gene flow ($N_m$) among populations was 1.082, indicating prevalent gene exchange *via* pollen dispersal. Phylogeographic analyses suggested that the populations of *G. kiangnanense* were divided into two lineages, Zhejiang (ZJ) and Anhui (AH). These two lineages were separated by the Huangshan-Tianmu Mountain Range. AMOVA analysis revealed that 36.59% of total genetic variation occurred between the two groups. The ZJ lineage was further divided into the Hangzhou (ZJH) and Zhuji (ZJZ) lineages, separated by the Longmen Mountain and Fuchun River. DIYABC analyses suggested that the ZJ and AH lineages were separated at 5.592 ka, likely due to the impact of Holocene climate change and human activities. Subsequently, the ZJZ lineage diverged from the ZJH lineage around 2.112 ka. Given the limited distribution of *G. kiangnanense* and the significant genetic differentiation among its lineages, both *in-situ* and *ex-situ* conservation strategies should be implemented to protect the germplasm resources of *G. kiangnanense*.

## INTRODUCTION

*Gymnospermium kiangnanense* (P. L. Chiu) Lecomte is a perennial herbaceous plant of Berberidaceae. The genus *Gymnospermium* Spach comprises approximately 11 species. These species are distributed across a wide range from southeastern Europe to East Asia, primarily in the northern temperate zone (*Rosati et al., 2019*; *Song et al., 2022*). *Gymnospermium kiangnanense* is the only species distributed in the subtropical region (*Chiu, 1980*). The distribution range of *G. kiangnanense* is extremely limited, mainly confined to the southern Anhui Province and the northern Zhejiang Province, growing at forest edges at an altitude of 100–800 m (Table 1). In our field survey, only eight populations were found across its distribution region (Fig. 1). The distribution area of all populations was less than 10,000 m$^2$. Plants of *G. kiangnanense* bloom in early spring with brightly yellow flowers. Their tubers are effective in clearing heat, removing toxins, and treating blood stasis (*Liao, Wang & Xiao, 2001*). Due to its high medicinal and ornamental values, *G. kiangnanense* has been overexploited for decades. Natural germplasm resources of *G. kiangnanense* are declining, and survival of the species is seriously threatened (*Yu et al., 2021*). *Gymnospermium kiangnanense* has been listed as a key protected plant by Zhejiang and Anhui provinces in eastern China in 2012 and 2022 (*Zhejiang Provincial People's Government, 2012*; *Anhui Provincial People's Government, 2022*).

Genetic diversity is a crucial component of biodiversity. It maintains the evolutionary potential of species and enables them to respond to environmental changes (*Bhattacharyya & Kumaria, 2015*). The genetic variation pattern of plant populations can reflect life cycle characteristics important for plant adaptation. Environmental and geographic factors can affect genetic diversity and variation. Studies have shown that rare or endemic species with limited range and small population sizes often exhibit lower genetic diversity compared to those with wide geographic distribution (*Chung et al., 2018*; *Zaya et al., 2017*). Within *Gymnospermium*, molecular genetics analyses have been conducted in two species, *Gymnospermium microrrhynchum* (S. Moore) Takht. and *Gymnospermium scipetarum* Paparisto & Qosja ex E. Mayer & Pulevic. These species are both rare or endemic (*Lee, Yeon & Shim, 2016*; *Marzario et al., 2022*). *G. microrrhynchum*, primarily found in the high-altitude Baekdudaegan mountain range in Korea, exhibited a relatively low level of genetic diversity, high genetic differentiation, and restricted gene flow. These patterns likely result from the isolation of populations in alpine areas during the Late Pleistocene interglacial phases (*Lee, Yeon & Shim, 2016*). Conversely, a low level of population genetic diversity and weak population differentiation was observed in *G. scipetarum* subsp. *eddae*, an Italian endemic species restricted to a narrow forest area in the southern Apennines (*Marzario et al., 2022*). The subpopulations of *G. scipetarum* subsp. *eddae* are scattered over a surface of only a few kilometers along the same mountain range, suggesting that the present-day distribution pattern could be the result of a relatively recent fragmentation process (*Marzario et al., 2022*).

**Table 1 Sampling sites and individual number of *G. kiangnanense* in this study.**

| Population | Location | Longitude (E) | Latitude (N) | Altitude (m) | Sample size |
|---|---|---|---|---|---|
| ZJZ1 | Banqiu Village, Huanshan Town, Zhuji Country, Zhejiang Province | 120.285701 | 29.449250 | 427 | 21 |
| ZJZ2 | Fenglinxia Village, Huanshan Town, Zhuji Country, Zhejiang Province | 120.240882 | 29.514709 | 565 | 22 |
| ZJH1 | Lingshan Mountain, Xihu District, Hangzhou Country, Zhejiang Province | 120.029198 | 30.117410 | 166 | 11 |
| ZJH2 | Heqiao Village, Lin'an District, Hangzhou Country, Zhejiang Province | 119.265286 | 30.112928 | 106 | 24 |
| ZJH3 | Tuankou Village, Lin'an District, Hangzhou Country, Zhejiang Province | 119.163841 | 30.046660 | 108 | 20 |
| ZJH4 | Linqi Village, Lin'an District, Hangzhou Country, Zhejiang Province | 119.130800 | 29.946153 | 273 | 20 |
| AH1 | Laoshan Mountain, Guichi District, Chizhou Country, Anhui Province | 117.747187 | 30.328034 | 729 | 10 |
| AH2 | Guniujiang, Qimen Country, Anhui Province | 117.644439 | 30.110121 | 132 | 15 |

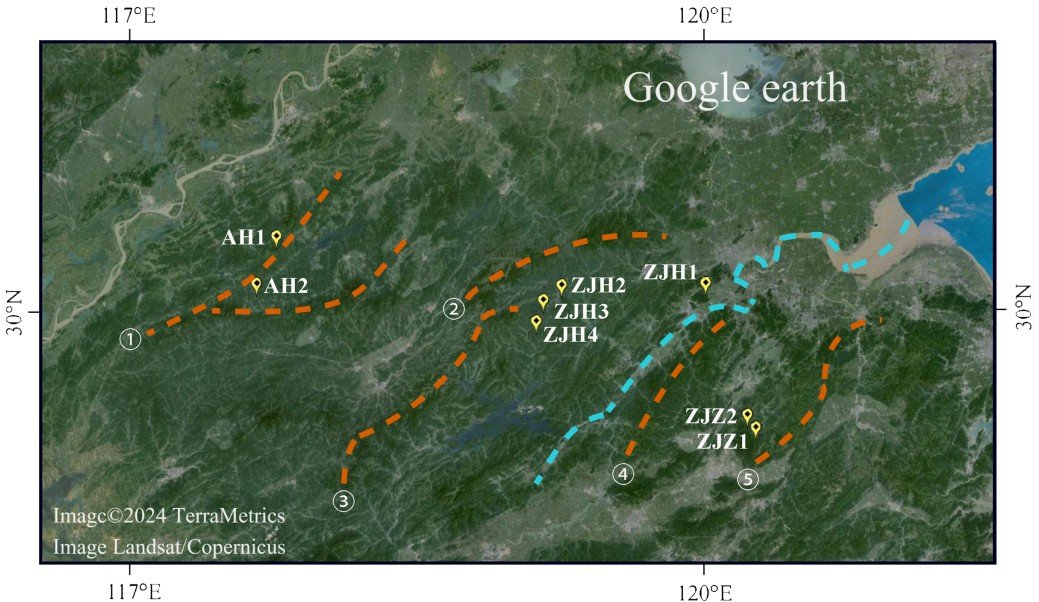

**Figure 1 Geographic locations of eight *G. kiangnanense* populations in this study.** Orange dotted lines indicate mountain range: ① Huangshan Mountain, ② Tianmu Mountain, ③ Qianligangs, ④ Longmen Mountain, ⑤ Kuaiji Mountain. The blue dotted line indicates the Fuchun River which separates the lineage of Zhejiang into two sublineages, ZJH and ZJZ. Map source: Google Earth.

Similar to *G. microrrhynchum* and *G. scipetarum* subsp. *eddae*, *G. kiangnanense* has a very narrow geographic distribution range. In the field survey, we found that it is limited to slope grasslands, forest understory, or gullies in the mountainous areas of Zhejiang and Anhui Provinces. So far, researches of *G. kiangnanense* have been focused on taxonomy, extraction of chemical components, and pharmacological mechanisms (*Chiu, 1980*; *Liu et al., 1992*; *Li et al., 1993*; *Liu, Rui & Deng, 1994*; *Liao, Wang & Xiao, 2001*). Few studies have explored the genetic resources of *G. kiangnanense*. Therefore, a population genetics study of *G. kiangnanense* is necessary for a better understanding of its endangered status and the development of effective conservation strategies.

Microsatellites, also known as simple repeats or short tandem repeats (SSR), consist of repeated units of 1–6 bases. Microsatellite sequences are widely distributed in plant genomes and are among the most informative and versatile genetic markers in plant functional genomics (*Taheri et al., 2018*). Microsatellites techniques have been widely used in population genetics studies of endangered species, such as *Sagittaria natans* Pall. (Alismataceae), *Davidia involucrata* Baill. (Nyssaceae), and *Toona ciliata* var. *pubescens* (Franch.) Hand. -Mazz. (Meliaceae) (*Yue et al., 2011*; *Ma et al., 2015*; *Liu, Jiang & Chen, 2014*).

Here we conducted population genetics study of *G. kiangnanense* based on SSR markers. The main objectives were to: (1) assess the level of genetic diversity, (2) examine the genetic structure and evaluate the genetic variation within and among populations, (3) infer the demographic history of the species, and (4) provide conservation implications for efficient protection of genetic resources and sustainable utilization of this species.

## MATERIALS AND METHODS

### Population sampling

The entire distribution area of Zhejiang and Anhui Provinces was surveyed from April 2016 to March 2021. Like most of extremely limited distributed species, *G. kiangnanense* usually have a population distribution range less than 10,000 m$^2$. In our field survey, we found that it is reproduced by seedling, without clones. We collected as many individuals as possible for each population. The collected individuals grow apart from each other for at least 10 m to avoid sampling of clones from the same maternal plant. A population generally contains 10–22 individuals. The sampling locations were recorded using a GPS locator (Table 1; Fig. 1). Leaves from each individual were sampled and placed into a tea bag with sampling code. Then, all the bags with leaves were dried with silica gel, and stored at −20 °C.

### Microsatellite marker development

Primers of polymorphic microsatellite loci were designed using the CandiSSR program based on transcriptome data of *G. kiangnanense* (GenBank accession numbers: SRR27230252, SRR27235853). The parameters were set as follows: flanking sequence length 100, comparison value $1 \times 10^{-10}$, comparison recognition 95, comparison coverage 95. Finally, SSR primers with ideal amplification efficiency and high polymorphism were selected for genetic analysis (Table S1).

### DNA extraction, PCR amplification and detection

Total genomic DNA was extracted using the Plant DNAzol (*Doyle & Doyle, 1987*) from approximately 0.1 g of silica-dried young leaf samples. The quantity of the obtained DNA was examined using a NanoDrop spectrophotometer, and PCR amplification products were detected using agarose gel electrophoresis. PCR amplifications were conducted in 15.0 ml reactions containing 30.0–100.0 ng genomic DNA, 2.0 mM of each primer, and 7.5 ml 2 × Taq PCR Master mix. The PCR program was as follows: Pre-denaturation at 95.0 °C for 3.0 min, denaturation at 94.0 °C for 2.0 min, annealing at 55.6–59.6 °C for 30.0

s, and extension at 70.0–75.0 °C for 10.0 min with a total of 30 cycles (Table S1). After completion of the amplification, the PCR products were run on a 1.5% agarose gel with the addition of 1.5 µL of EB staining. The gel was placed in 1 × TBE buffer and subjected to electrophoresis at 80 V for 20 min. All the electrophoresis images were shown in Fig. S1. DNA fragments were visualized under UV light and photographed using a Gel Documentation System (Bio-Rad, Hercules, CA, USA). The amplified PCR products were sent to Tsingke Biotech Company (Chengdu, Sichuan, China) for sequencing. The raw reads were shown in Table S1.

## Data analyses

### Genetic diversity and genetic structure

The Cervus 2.0 software (*Marshall et al., 1998*) was used to calculate the mean polymorphic information content ($PIC$), number of alleles ($N_A$), expected heterozygosity ($H_E$), observed heterozygosity ($H_O$), Wright's fixation index ($F_{ST}$), intrapopulation inbreeding coefficient ($F_{IS}$), and total population inbreeding coefficient ($F_{IT}$) for each locus. The linkage disequilibrium (LD) among 21 loci in this study were detected by GENEPOP version 4.7 (*Rousset, 2008*). The genetics indexes for each population were calculated by GenAlEX version 6.5 (*Peakall & Smouse, 2006*).

Poptree version 2.0 (*Takezaki, Nei & Tamura, 2010*) were used to constructed the phylogenetic UPGMA tree based on the pairwise genetic distances of the eight *G. kiangnanense* populations calculated by GenAlEx version 6.5 (*Peakall & Smouse, 2006*). Principal coordinate analysis (PCoA, *Peakall & Smouse, 2006*) of all the 143 *G. kiangnanense* individuals in this study was also conducted using GenAlEx version 6.5. Hierarchical analyses of molecular variance (AMOVA) within and between lineages or populations were assessed by Arlequin version 3.11 (*Excoffier, Laval & Schneider, 2005*). Bayesian clustering method in STRUCTURE version 2.3.4 (*Raj, Stephens & Pritchard, 2014*) was used to identify the geographical structure of the genetic variation in SSR sequences.

### Population divergence history

The population history of *G. kiangnanense* was investigated using the Approximate Bayesian Computation (ABC) approach implemented in DIYABC v.2.1 (*Cornuet et al., 2014*). Based on the results from STRUCTURE and systematic clustering analysis, the studied populations were divided into two major groups: populations from Zhejiang Province (ZJ) and Anhui Province (AH). The ZJ clade was further divided into two sub-clades: populations from Zhuji City (ZJZ) and Hangzhou City (ZJH). Three possible historical divergence scenarios were proposed for these three groups (Fig. 2).

In scenarios 1, populations of the AH group and ZJ group initially diverged from the common ancestor of the two lineages, followed by recent divergence of the ZJZ group from the ZJH group. In scenarios 2, populations of the AH group and ZJ group initially diverged from the two lineages, followed by recent divergence of the ZJH group from ZJZ group. In scenarios 3, populations from the ZJH, ZJZ and AH groups diverged simultaneously (Fig. 2).

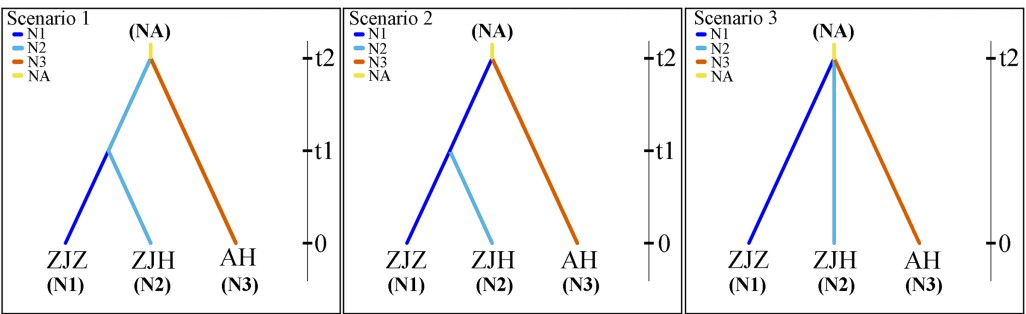

**Figure 2  Most likely evolutionary models of *G. kiangnanense*.**

To evaluate the optimum parameter values of the DIYABC model, the entire dataset is subjected to initial calculation under the assumption of uniform distribution. The minimum population size is set to 10, and the maximum is set to 100. The minimum population separation time is set to 10 generations, and the maximum is set to 10,000. The average mutation rate per generation for each locus was set to $5 \times 10^{-4}$ with a minimum value of $10^{-5}$ and maximum value of $10^{-2}$. A total of one million pseudo observation datasets (PODs) were simulated, then the operating parameters were adjusted and determined based on the 95% confidence interval of the simulated results until the optimal support plan is obtained. The posterior probability and corresponding 95% CI value for each scenario were calculated using a local linear regression program. The optimal calculating method was selected for principal component analysis (PCA) to check the fit degree between simulated values and observed values.

## RESULTS

### Primer polymorphism analyses

After screening for loci polymorphism, a total of 21 pairs of SSR primers were used in the study. Agarose gel electrophoresis images showed that the amplification products of the 21 pairs of SSR primers had clear bands (Fig. S1). The number of alleles per locus ranged from two to six with a mean value of 2.900. The expected heterozygosity ($H_E$) ranged from 0.007 to 0.762 with a mean value of 0.267. The observed heterozygosity ($H_O$) ranged from 0.007 to 0.979 with a mean value of 0.212 (Table S3). The *Ho* values were generally smaller than *He* values, indicating the presence of different sets of heterozygous deletions. The polymorphic information content (*PIC*) ranged from 0.007 to 0.721, with a mean value of 0.236. Among these, 12 loci showed low polymorphism ($\leq 0.25$), eight loci showed moderate polymorphism ($0.250 < PIC \leq 0.500$), and one loci showed highly polymorphism ($PIC > 0.500$). The total populations inbreeding coefficient ($F_{it}$) ranged from $-0.736$ to 0.866, with a mean value of 0.209. The genetic differentiation coefficient ($F_{st}$) among populations of *G. kiangnanense* ranged from 0.021 to 0.780, with a mean value of 0.261. The intra-population inbreeding coefficient ($F_{is}$) ranged from $-0.949$ to 0.839, with a mean value of $-0.050$. Among these, nine loci had heterozygous deletion, and the remaining

twelve loci had some heterozygous excess. The interpopulation gene flow ($N_m$) ranged from 0.071 to 11.491, with a mean value of 2.559.

## Genetic diversity of *G. kiangnanense*

The genetic diversity of eight *G. kiangnanense* populations covering its natural distribution areas was analyzed, revealing different degrees of genetic diversity (Table 2). The number of alleles ($N_A$) and effective alleles ($N_E$) ranged from 1.619 (ZJH1, ZJH2, ZJH4) to 1.952 (ZJZ1) and from 1.247 (ZJH3) to 1.445 (AH1), respectively. The mean value of $N_A$ was 1.714, and the mean value of $N_E$ was 1.330. The $H_O$ ranged from 0.172 (ZJZ1) to 0.300 (AH1), with a mean value of 0.222. The $H_E$ ranged from 0.143 (ZJH3) to 0.240 (AH1), with a mean value of 0.179. The $H_O$ of ZJZ1 was less than $H_E$, indicating a low genetic diversity in the population. The Shannon information index ($I$) ranged from 0.228 (ZJH3) to 0.366 (AH1), with a mean value of 0.286. The mean value of the fixed coefficient ($F$) for each population was −0.104 ($F < 0$). The mean value of the inbreeding coefficient ($F_{IS}$) for each population was −0.206 ($F_{IS} < 0$), indicating an unbalanced distribution of genotypes within the population, the presence of heterozygotes in each population, the presence of inbreeding in ZJZ1, and the existence of inbreeding and distant crosses within the remaining seven populations. The mean percentage of polymorphic loci (PPL) was 57.14%.

## Genetic structure of *G. kiangnanense*

In order to further investigate the genetic clustering relationship and genetic structure among populations of *G. kiangnanense*, principal coordinate analysis (PCoA) was conducted using GenAlEx. The results showed that the eight natural populations were divided into two groups (Fig. 3). UPGMA cluster analysis based on genetic distance using Poptree also supported the above results (Fig. 4).

STRUCTURE analysis yielded the highest likelihood when samples were clustered into two groups (K = 2; Figs. 5A and 5B). The majority of individuals (96.20%) from Zhejiang Province populations (ZJZ1, ZJZ2, ZJH1, ZJH2, ZJH3, and ZJH4) were assigned to cluster I ('light blue pool', Figs. 5 and 6), while individuals (97.90%) from Anhui Province populations (AH1 and AH2) were assigned to cluster II ('orange pool', Figs. 5 and 6). In addition, STRUCTURE analysis of the six *G. kiangnanense* populations in Zhejiang Province showed that K = 2 is the best grouping, with ZJZ1 and ZJZ2 from Zhuji City forming one lineage and ZJH1, ZJH2, ZJH3 and ZJH4 from Hangzhou City forming the other lineage (Figs. 5C and 5D).

## Genetic differentiation and genetic variation of *G. kiangnanense*

Population subdivisions into the Zhejiang (ZJ) and Anhui (AH) groups explained 36.59% of the total variation in AMOVA analyses, while 47.84% of the variation was within individuals. The overall $F_{ST}$ value was 0.521, indicating high genetic differentiation among groups (Table 3). We also examined the genetic differentiation among populations located in Zhejiang Province, which were further divided into two lineages, ZJZ and ZJH. AMOVA results showed an overall $F_{ST}$ value of 0.301, with 13.68% of the variation existed among groups and 69.87% occurred within groups (Table 3). Accordingly, the pairwise $F_{ST}$ values

**Table 2 Genetic diversity of *G. kiangnanense* populations based on SSR analysis using GenALEX versin 6.5.**

| Pop | $N$ | $N_A$ | $N_E$ | $H_O$ | $H_E$ | $I$ | $F$ | $F_{IS}$ | PPL |
|---|---|---|---|---|---|---|---|---|---|
| ZJZ1 | 21.000 | 1.952 | 1.314 | 0.172 | 0.179 | 0.310 | 0.201 | 0.060 | 71.43% |
| ZJZ2 | 22.000 | 1.762 | 1.276 | 0.188 | 0.153 | 0.254 | −0.054 | −0.195 | 57.14% |
| ZJH1 | 11.000 | 1.619 | 1.373 | 0.273 | 0.210 | 0.312 | −0.223 | −0.353 | 61.90% |
| ZJH2 | 24.000 | 1.619 | 1.331 | 0.220 | 0.177 | 0.273 | −0.167 | −0.222 | 47.62% |
| ZJH3 | 20.000 | 1.667 | 1.247 | 0.174 | 0.143 | 0.228 | −0.077 | −0.283 | 57.14% |
| ZJH4 | 20.000 | 1.619 | 1.315 | 0.217 | 0.158 | 0.255 | −0.249 | −0.208 | 52.38% |
| AH1 | 10.000 | 1.714 | 1.445 | 0.300 | 0.240 | 0.366 | −0.163 | −0.197 | 61.90% |
| AH2 | 15.000 | 1.762 | 1.339 | 0.229 | 0.174 | 0.293 | −0.194 | −0.255 | 52.38% |
| Mean | 17.875 | 1.714 | 1.330 | 0.222 | 0.179 | 0.286 | −0.104 | −0.206 | 57.14% |
| Total | 143.000 | 2.905 | 1.563 | 0.212 | 0.266 | 0.484 | 0.214 | −1.653 | 100.00% |

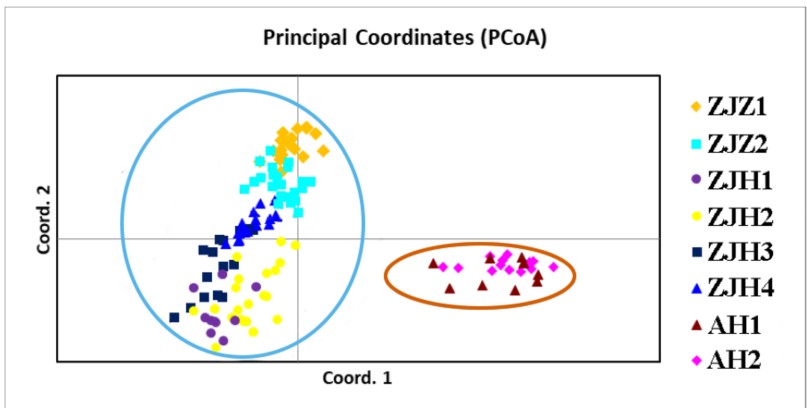

**Figure 3 Principle coordinate analysis (PCA) of eight *G. kiangnanense* populations based on genetic distance ($D_A$).** Populations from the same region are marked with the same color code (orange for Anhui, red for Hangzhou and yellow for Zhuji). The first and second axes extracted 58.83% and 18.50% of the total genetic variation, respectively.

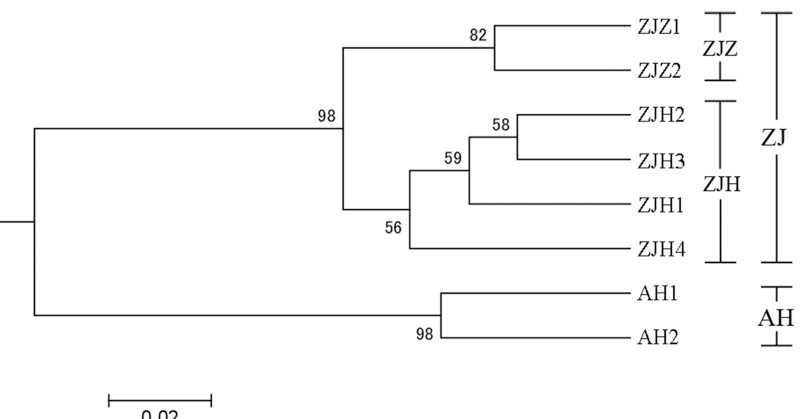

**Figure 4 UPGMA tree of eight *G. kiangnanense* populations based on 21 SSR loci.** Numbers on branches indicate bootstrap values from 5,000 replicates. (See Table 1 for population codes).

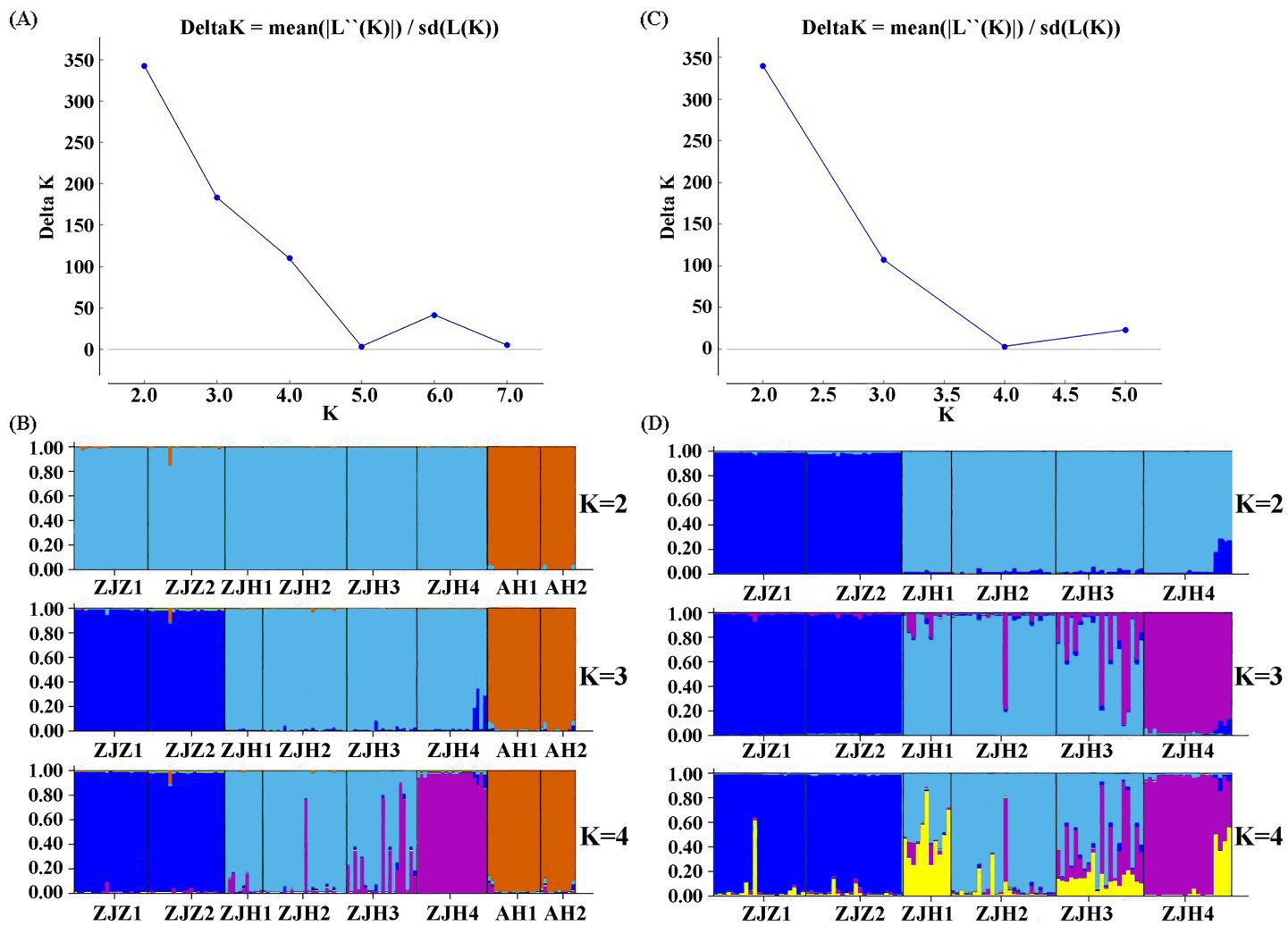

**Figure 5 Summary of structure analyses based on 21 SSR loci of *G. kiangnanense*.** (A) ΔK statistics of the genetic structure analysis of eight *G. kiangnanense* populations; (B) genetic structure from different K values of the eight *G. kiangnanense* populations. (C) ΔK statistics of the genetic structure analysis of the six populations from Zhejiang Province; (D) genetic structure from different K values of the six *G. kiangnanense* populations from Zhejiang Province.

among populations were significant, ranging from 0.062 to 0.122 (Table 4). Furthermore, the gene flow between the two lineages of *G. kiangnanense* was 2.792, lower than the gene flow between the two sub-populations, ZJZ and ZJH in Zhejiang Province, which is 4.323.

## Population divergence history

Based on results from genetic structure analysis and phylogenetic reconstruction, three hypothetical evolutionary scenarios were designed and further evaluated using DIYABC. The optimal model was determined by continuously correcting the parameter ranges, and the relative posteriori probabilities for each scenario were estimated using multivariate logistic regression methods and direct estimation methods in comparison with the observed dataset. The comparison between the posterior probabilities of the three scenarios using local linear regression showed that model 1 was the most likely scenario,
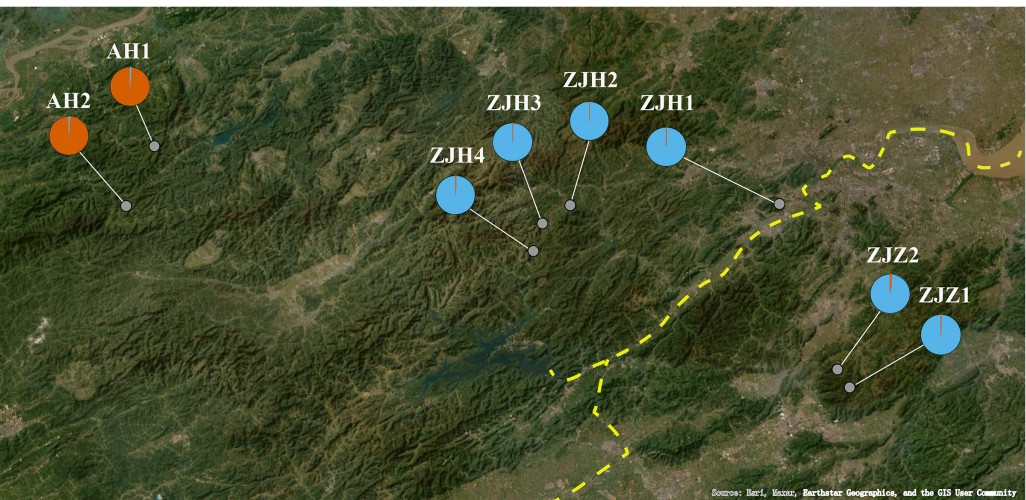

**Figure 6** **The geographical lineage structure _G. kiangnanense_ based on the STRUCTURE analyses.** The yellow dotted line indicates the Fuchun River.               

**Table 3** **Analyses of molecular variance (AMOVA) and the gene flows of the main lineages of _G. kiangnanense_.**

|  | Among groups | Among populations within groups | Within populations | $F_{ST}$ | $N_m$ |
|---|---|---|---|---|---|
| ZJ-AH | 36.59 | 15.57 | 47.84 | 0.521 | 2.792 |
| ZJZ-ZJH | 13.68 | 16.45 | 69.87 | 0.301 | 4.323 |

with a posterior probability of 0.570 (95% CL [0.530–0.610]), which is much higher than the model 2 (0.374; 95% CL [0.335–0.414]) and the model 3 (0.054; 95% CL [0.031–0.078]).

  According to the identified best scenario, the _G. kiangnanense_ populations from Anhui Province (AH, effective population size of ~617 individuals) most likely diverged firstly from the common ancestor of the group from Zhejiang and Anhui province around 932 generations ago (95% HPD [highest posterior density] 235.000 to 1,900.000; Table S4; Fig. S1). Within Zhejiang Province, divergence between populations from Zhuji (ZJZ, effective population size of ~1,170 individuals) and from Hangzhou (ZJH, effective population size of ~825 individuals) occurred around 352 generations ago (95% HPD [highest posterior density] 58.600 to 965.000; Table S4; Fig. S1). According to previous observation on the life cycle of _G. kiangnanense_, the species starts to bloom in about 6 years and bear fruits in the same year. Therefore, a 6-year generation time was assumed, which places the divergence between the AH and the common ancestor of ZJZ and ZJH populations at 5.592 ka (95% HPD [1.410–11.400 ka]), and the divergence between the ZJZ and ZJH populations at 2.112 ka (95% HPD [0.351–5.790 ka]). The changes in the number of effective populations in AH, ZJZ and ZJH may indicate an expansion trend of _G. kiangnanense_.

## DISCUSSION

### Genetic diversity of *G. kiangnanense*

Genetic diversity is an important indicator of a species' ability to adapt to environmental changes and is considered as important reference for the development of conservation strategies for endangered plants. This study sampled eight *G. kiangnanense* populations with a total of 143 individuals across the entire distribution range and obtained molecular data from 21 microsatellite polymorphic loci to examine the genetic diversity of *G. kiangnanense*. Cervus analysis showed that the polymorphic information content (*PIC*) of *G. kiangnanense* ranged from 0.007 to 0.721 with a mean value of 0.236, indicating the polymorphic nature of these loci. The average number of alleles ($N_A$) was 1.714, which was close to the average number of effective alleles ($N_E = 1.330$), indicating that the alleles of the 21 loci used in this study were evenly distributed.

The expected heterozygosity ($H_E$) values of the eight populations of *G. kiangnanense* ranged from 0.143 to 0.240 with a mean value of 0.179, indicating a relatively low genetic diversity. However, genetic diversity at the species level was relatively high ($H_E = 0.266$). This result is consistent with *Lee, Yeon & Shim (2016)*, who reported that seven populations of *G. microrrhynchum* distributed in Manchuria of Korean Peninsula had low genetic diversity and relatively high genetic diversity at the species level. The study of five *G. scipetarum* populations distributed in Italy using RAPD genetic markers also showed relatively low levels of genetic diversity (*Marzario et al., 2022*). The average $H_E$ values of *G. kiangnanense*, *G. microrrhynchum*, and *G. scipetarum* subsp. *eddae* populations were very close (*G. microrrhynchum* = 0.188; *G. eddae* = 0.187; *G. kiangnanense* = 0.179).

The genetic diversity of plant populations is influenced by factors such as the extent of regional distribution, age and size, breeding system, seed dispersal, and life history (*Hamrick, Godt & Sherman-Broyles, 1992*). Similar to G. *kiangnanense*, some other endangered plants exclusively or partially distributed in Zhejiang Province, such as *Sinojackia microcarpa* C.T. Chen & G.Y. Li and *Heptacodium miconioides* Rehder, also exhibit low levels of population genetic diversity (*Zhong et al., 2019*; *Li & Jin, 2006*). A low value of Shannon information index (*I*) indicated lower inter-population biodiversity and the geographical proximity of the eight *G. kiangnanense* populations. Additionally, *Yu et al. (2021)* observed a declining age structure in all populations of *G. kiangnanense*, indicating a decline in the breeding system. *G. kiangnanense* is heterogamous, depending on pollinating insects for reproduction, yet it exhibits a low seed setting rate and weak seed dispersal ability. Consequently, the species faces challenges in reproduction due to its narrow regional distribution, limited seed dispersal distance, and potentially increasing anthropogenic disturbances. These factors collectively pose serious threats to the survival of *G. kiangnanense*, thereby impacting species diversity.

### Genetic differentiation and structure of *G. kiangnanense*

Results from AMOVA analysis revealed a high level of population differentiation and a significant population structure among all the populations ($F_{ST} = 0.521$). The mean value of gene flow ($N_m$) among populations was 2.559. *Gymnospermium kiangnanense* is an

entomophilous plant, primarily relying on insects such as *Apis cerana* Fabricius, *Episyrphus balteatus* De Geer, *Scaeva pyrastri* Linnaeus, and *Halictus* sp. for pollination (*Yu et al., 2021*). Considering the abundance of pollen grains and the ease of pollen dispersal, gene flow *via* pollen may be more prevalent than that *via* seed, resulting in a relatively lower ratio of genetic variation among populations within groups and a low degree of habitat fragmentation.

The genetic structure of a species is crucial for evaluating its genetic resources. Analysis of genetic differentiation patterns can provide valuable insights for the effective conservation and utilization of these resources. Our analyses, including PCoA, UPGMA, and STRUCTURE, consistently identified two major genetic groups within *G. kiangnanense*. The eastern populations in Zhejiang Province formed one distinct lineage (ZJ), while the western populations in Anhui Province comprised the other lineage (AH) (Figs. 3–5). The hierarchical AMOVA analysis indicated a high level of genetic differentiation ($F_{ST} = 0.521$), with 36.59% of the variation occurs between populations in the Zhejiang (ZJ) and Anhui (AH) groups (Table 3). Within Zhejiang Province, the six *G. kiangnanense* populations were further divided into two lineages. The southeast populations in Zhuji City formed one lineage (ZJZ), while the northwest populations in Hangzhou City comprised the other lineage (ZJH). The $F_{ST}$ value between the ZJZ and ZJH lineages was 0.301, supporting a significant genetic structure within Zhejiang Province (Table 4).

Due to the complex effects of mountains, rivers, and climate in Southeast China, the geographic region in Southeast China displays high levels of ecosystem and species diversity. It harbors a large number of endemics, endangered, and tertiary relict species. The fruit of *G. kiangnanense* lacks a seed dispersal vector and relies solely on gravity for dispersal. Consequently, seed dispersal can be easily hindered by mountain slopes, streams, and rivers. Therefore, as one of the main mountain ranges in East China, the Huangshan-Tianmu Mountain Range located on the boundary of Zhejiang and Anhui Provinces may serve as a natural boundary separating populations of *G. kiangnanense*. Within Zhejiang Province, populations distributed in the northwestern part and the remnants of Tianmu Mountain were clustered together, while populations distributed between the Kuaiji and Longmen Mountain Range formed a different lineage (Fig. 1). The Longmen Mountain and Fuchun River may have contributed to the genetic differentiation between the two lineages (Fig. 1). Overall, the phylogeographic pattern of *G. kiangnanense* suggests a vicariant event that divided an ancestral stock into different sub-groups. The barrier generally conforms to the geographical boundaries caused by mountains and water systems in Zhejiang and Anhui Provinces.

## Demographic history

The DIYABC analysis supported Scenario 1, indicating that the AH lineage of *G. kiangnanense* first separated from the common ancestor of current populations. The divergence time (t2) was predicted as 5,592 years. Such demographic events may be linked to post-Holocene climate change and human activities. Similar to *G. microrrhynchum*, *G. kiangnanense* is found in temperature-sensitive habitats. It is presumed to start
**Table 4 Analysis of genetic differentiation ($F_{ST}$) of the eight populations and the gene flows of *G. kiangnanense*.**

| Population | $F_{ST}$ | $N_m$ |
|---|---|---|
| AH1–AH2 | 0.113 | 1.041 |
| ZJZ1–ZJZ2 | 0.062 | 1.264 |
| ZJH1–ZJH2 | 0.096 | 1.177 |
| ZJH1–ZJH3 | 0.070 | 1.408 |
| ZJH1–ZJH4 | 0.122 | 0.639 |
| ZJH2–ZJH3 | 0.070 | 1.458 |
| ZJH2–ZJH4 | 0.118 | 0.706 |
| ZJH3–ZJH4 | 0.075 | 0.963 |
| Average | 0.091 | 1.082 |

migration to subtropical regions during the middle Miocene (15.680 Mya) when the global climate became cold and arid, and was thus diverged from its close relatives at around 14.670 Mya (*Song et al., 2022*). During the middle Holocene (8.450–2.450 ka), the weather became warm and humid, with a significant increase in summer precipitation, exceeding 0.500 mm/d (*Zheng et al., 2004*). These warmer and wetter conditions, coupled with gradually increased human activities, triggered the growth of terrestrial herbs and the gradual opening of the original evergreen and deciduous mixed broad-leaved forest between ca. 7.550 and 3.750 ka, as supported by archaeological evidence (*Chen, Wang & Dai, 2009*). Hence, the restricted distribution of *G. kiangnanense* in mountainous areas and its high inter-group genetic differentiation may be due to its inefficient seed dissemination, as well as its inability to cope with climatic amelioration and compete with other herbaceous plants under warm conditions and human interference. Examples of human activities affecting population structure can also be observed in *Parrotia subaequalis* (H. T. Chang, R. M. Hao & H. T. Wei) (Hamamelidaceae). A study of the species proposed that declined forests due to farmland expansion over the last 6,000 years have disrupted recent population connectivity (*Zhang et al., 2018*).

In the following 4,000 years, the climate fluctuated between warm and cold until 221 B.C. However, the impact of human activities during this period on natural vegetation has exceeded the influence of climate in some areas, even became the main driving force of vegetation change (*Boivin et al., 2016*). The broad-leaved forest disappeared after 3.750 ka, and anthropogenic deforestation and farmland management provided more space for herbaceous plants (*Chen, Wang & Dai, 2009*; *Wang et al., 2023*; *Ke et al., 2023*). It is hypothesized that the ZJZ lineage diverged from ZJH at 2.112 ka. Overall, our results indicate that climate change and human activities have significant impacts on the current genetic variation pattern of *G. kiangnanense* populations.

## CONCLUSIONS

Given the rarity of *G. kiangnanense* and its significant genetic differentiation among different lineages, it is necessary to establish nature reserves in the distribution range of
each lineage, in order to preserve the genetic diversity of both Zhejiang and Anhui populations. Since relatively high genetic variation was observed among different individuals within population, the disappearance of any individual may lead to a reduction in the genetic diversity of *G. kiangnanense*. Therefore, instead of focusing on a particular population, all *G. kiangnanense* populations need to be protected. Meanwhile, through the study of population breeding systems and simulation of suitable distribution areas, the potential distribution area for small population expansion can be identified. The future dynamic change and propagation route of *G. kiangnanense* populations can also be predicted. This information can help establish restricted-access conservation areas in buffering zones and predicted distribution areas to prevent habitat fragmentation and destruction caused by human intervention. Additionally, establishing an *ex-situ* germplasm bank along with seedling cultivation techniques may serve as practical methods for the conservation of *G. kiangnanense* resources.

In conclusion, this study first analyzed the genetic diversity of eight *G. kiangnanense* populations using 21 SSR loci and deduced historical demographic events using DIYABC. Our results showed that the eight populations were classified into two major groups in Zhejiang and Anhui Provinces, each harboring a unique gene pool. The Zhejiang populations were further divided into two sub-clades located on the opposite site of the Fuchun River and Longmen Mountain. The natural geographic boundaries and seed dispersal mechanism of *G. kiangnanense* may have contributed to the divergence of its populations. Holocene climatic changes and human activities likely further affected the current distribution pattern of the species. This study sheds light on the current status of genetic resources of *G. kiangnanense*, provides a scientific basis for understanding its population demographic history, and offers insights for effective conservation and sustainable utilization of this endangered species.

## ACKNOWLEDGEMENTS

We are grateful to Pan Li, Hongwei Zhang, Ruisen Lu, Guohua Xia, Hongliang Chen, Chaoqian Ren, Chenxi Wang and Shenglu Zhang for their great help with plant samples.

### Funding

This study was supported by the National Natural Science Foundation of China (No. 32060053) and the Zhejiang Provincial Public Welfare Technology and Application Research Project (No. LGN21C020007). Chenshan Special Foundations from Shanghai Municipal Administration of Forestation and City Appearances (G222404). The funders had no role in study design, data collection and analysis, decision to publish, or preparation of the manuscript.

### Grant Disclosures

The following grant information was disclosed by the authors:
National Natural Science Foundation of China: 32060053.

Zhejiang Provincial Public Welfare Technology and Application Research Project: LGN21C020007.

Shanghai Municipal Administration of Forestation and City Appearances: G222404.

## Competing Interests

The authors declare that they have no competing interests.

## Author Contributions

- Xiangnan Liu performed the experiments, analyzed the data, prepared figures and/or tables, authored or reviewed drafts of the article, and approved the final draft.
- Meizhen Wang performed the experiments, prepared figures and/or tables, authored or reviewed drafts of the article, and approved the final draft.
- Shiqiang Song performed the experiments, authored or reviewed drafts of the article, and approved the final draft.
- Qing Ma analyzed the data, authored or reviewed drafts of the article, and approved the final draft.
- Zhaoping Yang conceived and designed the experiments, analyzed the data, authored or reviewed drafts of the article, and approved the final draft.

## Data Availability

The raw measurements are available in Table 1 and Table S1.

## Supplemental Information

Supplemental information for this article can be found online at http://dx.doi.org/10.7717/peerj.17554#supplemental-information.

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
