# Peer review of "Population structure and diversification of Gymnospermium kiangnanense, a plant species with extremely small populations endemic to eastern China"

_PeerJ, doi:10.7717/peerj.17554_

## Round 0.1 · original submission · Major Revisions

Dear author your article is not acceptable in its current form. Please revise the article considering the reviewers' comments.

**Language Note:** The review process has identified that the English language must be improved. PeerJ can provide language editing services - please contact us at [email protected] for pricing (be sure to provide your manuscript number and title). Alternatively, you should make your own arrangements to improve the language quality and provide details in your response letter. – PeerJ Staff

Reviewer 1 ·

Basic reporting

This manuscript investigated the genetic diversity of Gymnospermium kiangnanense, an endangered species endemic to eastern China, based on 22 SSRs markers. The distribution of the species is largely limited in two provinces, though the species demonstrated strong population structure and low diversity within populations. This study provided their unique contribution to the conservation of these endangered species. Here I provided some suggestions.
1) Many sentences should be added with references, such as in the abstract, “Gymnospermium kiangnanense is an endangered species endemic to eastern China”. However, I do not see any references for this sentence in the introduction, I mean, is the species listed as endangered species in IUCN read list or national, provincial key protection species list? In the introduction part (line 59-62) “Due to its high medicinal and ornamental values, G. kiangnanense has been overexploited for decades. Natural germplasm resources of G. kiangnanense are declining and species survival is seriously threatened”. Also, no references were provided.
2) Table 3 could be deleted or treated as supplementary Table.
3) Table 5 is strange, please treated as two separate tables.
4) Figure 2: Is their any different meanings between scenario 1 and 2? The scenario in which ZJH diverged from AH first should also be tested. And gene flow between these lineages should also be tested.
5) Figure 3: please provide PCA analysis for all the individuals, it could be better.
6) Figure 5D could be deleted as it repeated Figure 5B.

Experimental design

no comment

Validity of the findings

no comment

Additional comments

no comment

Reviewer 2 ·

Basic reporting

My comments on introduction:
Clarity: Some sentences are lengthy and could be simplified for better readability. For example, sentences such as "Genetic diversity is an important component of biodiversity and is crucial for maintaining the evolutionary potential of species and their ability to respond to environmental change" could be broken down into shorter, clearer statements.

Transition: The transition between the discussion of species' characteristics and the introduction of the study objectives could be smoother. Consider adding a sentence or phrase to bridge the gap between these two sections and provide a clearer transition to the research objectives.

Conciseness: While it is important to provide background information, make sure that every detail directly contributes to the understanding of the study. Some information, such as the specific alkaloid constituents of the plant and their use in traditional medicine, may not be directly relevant to the genetic analysis conducted in the study and can be shortened or omitted.

Experimental design

the methods section provides a comprehensive and detailed description of the procedures followed in the study, enabling replication by other researcher

Validity of the findings

1) can you provide information about the SSR repeats found, such as exonic, intronic, 3' or 5' distribution of the identified repeats? for more information refer this reference (Genic microsatellite marker characterization and development in little millet (Panicum sumatrense) using transcriptome sequencing)
2) authors are requested to add gel images as supplementary data
3)Discussion of experimental findings and analyses may be added.

Additional comments

The manuscript is scientifically written in a detailed manner and the research subject is attractive. However, before acceptance, the manuscript should be checked for technical errors, grammatical, and other typographical errors. The references must be rechecked.

Reviewer 3 ·

Basic reporting

no comment

Experimental design

no comment

Validity of the findings

no comment

Additional comments

The author conducted a survey and sampling of the entire distribution range of Gymnospermium kiangnanense and analyzed the genetic structure of eight populations using SSR. They evaluated the level of genetic diversity and assessed genetic variation within and among populations. Investigating the genetic resource conservation and sustainable utilization of Gymnospermium kiangnanense is of significant reference value. However, the author needs to pay attention to the following suggestions:

1. At the end of the introduction, the author should explicitly state the specific problems to be addressed, especially regarding the estimation of the differentiation time of different lineages of Gymnospermium kiangnanense, and what issues can be addressed in this paper (lines 102-106).

2. There is insufficient information provided about the natural distribution status of Gymnospermium kiangnanense. More detailed introductions are needed.

3. Please put the authorship when the species mentioned for the first time. Please also apply to all species. (e.g., line 74).

4. The introduction to sampling in the article is too simplistic; it is recommended to provide more detailed information, particularly regarding the justification for the sample size. (lines 110-113).

5. Decimal points in numerical results in the article should be consistent (e.g., lines 185-198).

6. The description of results in lines 246-248 needs to be supplemented with information about the data source.

---

## Round 0.2 · accepted · Accept

Revised article recommended for publication.

Reviewer 2 ·

Basic reporting

The revised manuscript has undergone significant improvements and has reached a level of quality suitable for publication in your esteemed Journal.

Experimental design

The revised manuscript has undergone significant improvements and has reached a level of quality suitable for publication in your esteemed Journal.

Validity of the findings

The revised manuscript has undergone significant improvements and has reached a level of quality suitable for publication in your esteemed Journal.

Additional comments

The revised manuscript has undergone significant improvements and has reached a level of quality suitable for publication in your esteemed Journal.